# Venous Thromboembolic Disease in Chronic Inflammatory Lung Diseases: Knowns and Unknowns

**DOI:** 10.3390/jcm10102061

**Published:** 2021-05-11

**Authors:** George Keramidas, Konstantinos I. Gourgoulianis, Ourania S. Kotsiou

**Affiliations:** 1Department of Respiratory Medicine, Faculty of Medicine, University of Thessaly, BIOPOLIS, 41110 Larissa, Greece; gkeramidas10@gmail.com (G.K.); kgourg@med.uth.gr (K.I.G.); 2Faculty of Nursing, School of Health Sciences, University of Thessaly, GAIOPOLIS, 41110 Larissa, Greece

**Keywords:** immunothrombosis, in situ thrombosis, lung diseases, pulmonary embolism, venous thromboembolism

## Abstract

Persistent inflammation within the respiratory tract underlies the pathogenesis of numerous chronic pulmonary diseases. There is evidence supporting that chronic lung diseases are associated with a higher risk of venous thromboembolism (VTE). However, the relationship between lung diseases and/or lung function with VTE is unclear. Understanding the role of chronic lung inflammation as a predisposing factor for VTE may help determine the optimal management and aid in the development of future preventative strategies. We aimed to provide an overview of the relationship between the most common chronic inflammatory lung diseases and VTE. Asthma, chronic obstructive pulmonary disease, interstitial lung diseases, or tuberculosis increase the VTE risk, especially pulmonary embolism (PE), compared to the general population. However, high suspicion is needed to diagnose a thrombotic event early as the clinical presentation inevitably overlaps with respiratory disorders. PE risk increases with disease severity and exacerbations. Hence, hospitalized patients should be considered for thromboprophylaxis administration. Conversely, all VTE patients should be asked for lung comorbidities before determining anticoagulant therapy duration, as those patients are at increased risk of recurrent PE episodes rather than DVT. Further research is needed to understand the underlying pathophysiology of in-situ thrombosis in those patients.

## 1. Introduction

Pulmonary embolism (PE) and deep vein thrombosis (DVT) are clotting disorders collectively named venous thromboembolism (VTE), constituting the third most frequent acute cardiovascular syndrome [1]. PE most frequently occurs as a result of DVT of the lower extremities. PE’s annual incidence ranges between 39 and 115 per 100,000 of the population [1]. Annual incidence for DVT ranges from 53 to 162 per 100,000 population [1]. Many studies support the notion that the VTE incidence showed a clear upward trend since the end of the 20th century when computed tomography pulmonary angiogram (CTPA) was introduced as an alternative to ventilation/perfusion scanning (V/Q scan) [2]. On the other hand, mortality seems to have a decreasing tendency since then at about 6,5 deaths per 100,000 population in 2015, according to data from the World Health Organization (WHO) Mortality Database [3].

Although several risk factors result in VTE, no risk factor can be identified in a third to half of VTE events [4], VTE is categorized as unprovoked in these cases, determining the duration of the anticoagulation therapy. Patients with no identifiable provoking factors for VTE may have individual risk factors that are rarely sufficient by themselves to predispose to a VTE episode. However, combinations thereof should be carefully reviewed to assess the overall risk of VTE in each patient.

Despite the close links and common evolutionary origin of inflammation and thrombosis, these pathways are mainly viewed separately [5]. However, inflammation and coagulopathy are tightly connected. Emerging evidence suggests that chronic inflammation may be a cause, as well as a consequence of VTE. The underlying mechanisms associated with chronic inflammation-associated coagulopathy have been suggested but not clearly defined.

It is helpful to conceptualize inflammation and thrombosis as interconnected not only from a basic science perspective but also from a clinical perspective [5]. Pulmonary thrombosis may not always result from embolization of a peripheral clot to the pulmonary arteries. Immunothrombosis normally facilitates the entrapment and disposal of pathogens and cellular debris. It is a phenomenon triggered by pathogens or cellular debris in inflammation, leading to thrombus formation [6]. However, once getting beyond control, it contributes to the formation of thrombus in situ. Evidence supports that patients with lung inflammatory diseases are at great risk of thrombosis directly in the lung’s vasculature. In these cases, there is an activation of endothelial cells, platelets, and leukocytes with subsequent microparticles forming that can trigger the coagulation system through the induction of tissue factor (TF) [7]. Clot formation also results from the excretion of neutrophil extracellular traps (NETs) by neutrophils. NETs can activate factor (F) VII and TF, bind to von Willebrand factor (vWF), activate platelets via H3 and H4 histones, and locally concentrate enzymes such as neutrophil elastase and myeloperoxidase which cleave and inactivate TF pathway inhibitor (TFPI) and thrombomodulin [6], triggering immunothrombosis. Moreover, through largely overlapping pathways, and with the possible additional mediation of agent-specific virulence factors or autoantibodies, infection by both viruses [8] and bacteria [9], to which patients with chronic and acute respiratory disease are particularly vulnerable, has been postulated to cause local or systemic thrombophlebitis.

Transendothelial migration, an increase in the expression of endothelial adhesion molecules and in the percentage of senescent endothelial cells, vascular atrophy due to endothelial apoptosis, vascular remodeling and dysfunctional endothelial tissue are upregulated in chronic obstructive lung diseases depending on the phenotype of the patient [10,11,12]. These are likely to play a role in the increased influx of inflammatory cells seen in these chronic obstructive lung diseases [10,11,12,13,14], constituting parallel pathogenetic mechanisms involved in de novo VTE [10,11,12]. Moreover, the endothelial-to-mesenchymal transition is a precursor to IPF and vascular obliteration [14,15], as this change of endothelial phenotype may increase the vascular responses to thrombosis, decrease permeability control, and increase fibrotic reactions [14,15].

Persistent inflammation within the respiratory tract underlies the pathogenesis of numerous chronic pulmonary diseases [16]. However, the relationship between lung diseases and/or lung function with VTE is unclear. Understanding the role of chronic lung inflammation as a predisposing factor for VTE may not only help determine the optimal management, but may also aid in the development of future preventative strategies. This review aimed to provide an overview of the available evidence regarding the relationship between the most common chronic inflammatory lung diseases with VTE.

## 2. Asthma and VTE

Asthma is a chronic inflammatory airway disease [17]. The reversibility of airway limitation and inflammation in asthma patients has long been challenged. Asthma has been associated with enhanced procoagulant and antifibrinolytic activities in the airways [18,19,20,21,22], which derive from TF overexpression mechanisms, reduced anticoagulant protein C system activity, and inhibition of fibrinolysis through over-production of plasminogen activator inhibitor type 1 (PAI-1).

A great deal of previous research has focused on the association between asthma and PE. Majoor et al. reported that severe asthma is an independent risk factor for PE, but not for DVT [23] (Table 1). However, this concept has been challenged by several studies reporting an increased risk of both PE and DVT in asthma patients, a discrepancy in results likely related to different sample sizes [24]. More recently, Zöller et al. [24] studied 114,366 Swedish-born patients with a first hospital diagnosis of PE, 76,494 patients with DVT and 6854 patients with both PE and DVT in comparison with age-, gender-and educational attainment-matched controls for each case, between 1981 and 2010. The authors showed that asthma increases the risk for PE, isolated DVT, or both [24]. Specifically, it was reported that the adjusted odds ratio (OR) was 1.4 (95% CI 1.4–1.5) for PE, 1.6 (95% CI 1.5–1.7) for DVT, and 1.6 (95% CI 1.3–2.0) for both entities in asthma patients compared to healthy individuals (Table 1). Earlier, a nationwide cohort study by Chung et al. [9] showed that the overall incidence rate of PE was three times higher in 31,356 patients with newly diagnosed asthma compared to 125,157 non-asthmatic individuals (10.2 versus 3.1 per 100,000 person-years). The overall hazard ratio (HR) of PE in asthmatic patients was 3.2 (95% CI 1.7–6.0), after adjusting for sex, age, comorbidities, and estrogen use (Table 1).

Interestingly, the risk for thromboembolism increased with a recent history of a diagnosis of asthma [24]. According to Chung et al., the adjusted HR for PE (after adjusting for sex, age, comorbidities, and estrogen use) was 3.4 (95% CI 1.7–6.7) during the first five years after diagnosing asthma [17] (Table 1).

A gender-related difference in PE prevalence has been reported, with males having a 3.2-fold increased risk of PE than females [17]. Asthmatic patients were at increased risk for PE across all age groups than non-asthmatic individuals, which increased even further with age increase [17]. Contradictory data are also presented showing that the OR declined with the increase of age [24]. In particular, it has been found that the OR for PE and DVT decreased from 2.2 (95% CI 1.9–2.6) and 2.0 (95% CI 1.7–2.4) for patients younger than 50 years of age to 1.5 (95% CI 1.4–1.6) and 1.6 (95% 1.4–1.7) for patients older than 75, respectively [24], but no age-specific statistical interaction in the group with PE and DVT was found. This finding is intriguing, considering that age is the strongest independent risk factor for VTE in the general population. Potential sources of bias for the study were that higher than the 91% of the VTE study group were older than 50 years old, 2.3% and 2.4% of the PE and DVT groups, respectively had asthma but with incomplete measures of asthma, as reported, mild asthma was not recorded, while the presence of residual and unmeasured confounding factors could not be excluded [24]. The same study reported more PE cases in older ages and that DVT was relatively more common at younger ages, but asymptomatic PE frequently occurs among DVT cases and asymptomatic DVT frequently occurs in PE cases [24]. In addition, Majoor et al. [23] reported a nine-fold or a 3.5-fold increased risk of PE in patients with severe asthma or mild-moderate asthma compared to the general population, respectively. Interestingly, the use of systemic corticosteroids seemed to increase the risk of VTE, as stated by Johannesdottir et al. [25] and Stuijver et al. [26]. Because these effects were adjusted for indicators of severity of underlying disease [25], or for age; sex; hospitalizations (for malignancy, pregnancy, trauma, surgery, inflammatory bowel disease, inflammatory athritides, chronic respiratory failure); and use of anticoagulants, drugs for respiratory diseases, antibiotics, antidepressants, hormone replacement therapy, and oral contraceptives within 90 d prior to the index date [26], and existed also for noninflammatory conditions [25], these results are unlikely to be explained by confounding by indication.

According to the study of Alzghoul et al. [27], the prevalence of PE was 19.1% in patients during an asthma exacerbation. An unambiguous relationship between the frequency of emergency department visits because of asthma exacerbations and PE incidence has been stated [24]. Chung et al. supported that the adjusted HR for PE increased from 1.0 (95% CI 0.3–2.9) for those having two or fewer visits to 9.0 (95% 4.3–18.9) for those with four or more visits [9]. Similarly, Zöller et al. found that the OR for PE shaping from 1.3 (95% CI 1.3–1.4) for one to two hospital admissions to 1.4 (95% CI 1.3–1.6) for three or four admissions and 1.82 for more than four hospital admissions [24]. However, it is unclear whether PE triggers bronchospasm, rendering itself the causative factor of the exacerbation.

Besides, the asthma patients diagnosed with PE had a longer hospitalization period and were transferred more frequently to the intensive care unit (ICU) than asthma patients without PE, while the mortality rate of asthma exacerbation patients with an acute PE has been reported to be 4% [27]. Several risk factors associated with the development of an acute PE during an asthma exacerbation have been recognized, including the following in descending order: a previous history of PE, the high CHA2DS2-VASc score on admission that estimates the bleeding and thrombotic risk, the simultaneous presence of hyperlipidemia, a previous history of DVT, the presence of cancer, the previous use of systemic corticosteroids, the obesity, and the medical history of atrial fibrillation [27]. Based on these factors, a predictive score for PE during an asthma exacerbation has been recently developed with an average predictive accuracy reaching 88% [27]. Asthmatic patients with diabetes did not show an increased risk for PE [17].

Sneeboer et al. [28] compared coagulation parameters between healthy subjects and mild, severe or prednisolone-dependent asthma patients. The study showed significantly higher levels of endogenous thrombin potential (ETP), plasmin-antiplasmin complexes (PAPc), PAI-1, and vWF in the peripheral blood of asthmatic patients compared to healthy subjects. The prothrombotic state was even more intense with increasing asthma severity. A positive correlation between ETP or vWF and peripheral blood neutrophil counts was found, but not with eosinophilic counts.

In conclusion, several studies showed that asthma increases the risk for PE. Gender and age-related differences in PE prevalence have been reported. PE incidence is linked to asthma severity. Asthma patients with PE had a longer hospitalization and ICU stay period. However, the pathophysiological mechanisms are still not yet fully understood. It is considered that inflammation plays a major role by altering the balance between procoagulant and fibrinolytic agents. The prothrombotic state is related to increasing asthma severity, the number of exacerbations and hospitalizations, and the longer duration of hospitalization.

## 3. Chronic Obstructive Pulmonary Disease and VTE

COPD is characterized by persistent respiratory symptoms and airflow limitation due to airway and/or alveolar abnormalities caused by noxious gases or particles, influenced by host factors like abnormal lung development [29]. The airflow limitation occurs by a mixture of small airway disease and lung parenchyma destruction [29]. Respiratory infections cause about 50–70% of COPD exacerbations, 10% percent are caused by environmental pollution, and up to 30% of COPD exacerbations have no clear etiology [30].

People with poor lung function are at increased atherothrombotic risk. COPD, even in the stable phase, is considered to be an independent risk factor for PE. Systemic inflammation is the main atherothrombotic abnormality in COPD, but hypoxia-related platelet activation, pro-coagulant status, and oxidative stress may play a role [31,32,33]. Interestingly, Factor V Leiden homozygosity has been previously associated with severe dyspnea and decreased pulmonary function [33]. After adjustment for VTE risk factors, it has been reported that obstructive spirometric patterns (HR 1.3, 95% CI 1.1–1.7), and not restrictive spirometric patterns (HR 1.2, 95% CI 0.8–1.6), were associated with an increased risk of VTE. These results provided by a prospective cohort study by Kubota et al. included projecting 22 to 24 years risk from baseline spirometry diagnosis [32]. In contrast, other studies supported atherothrombosis in people with restrictive spirometric dysfunction that is attributed to systemic inflammation [33].

Similar to asthma, chronic inflammation contributes to the pathogenesis of COPD. An increase of C-reactive protein, fibrinogen, interleukin (IL)-6, IL-8 and tumor necrosis factor-a, hypoxemia, oxidative stress, and endothelial dysfunction may alter the coagulation profile by increasing PAI-1 and decreasing prostacyclin release [34].

VTE risk seems to be elevated in patients with acute exacerbation (AE) of COPD. A previous systematic review and meta-analysis by Rizkallah et al. found an estimated prevalence of PE of 19.9% (95% CI 6.7–33.0) among patients who did and did not require hospitalization, while they proposed that one of four COPD patients who require hospitalization for an acute exacerbation might have PE [35] (Table 2). Lately, Pourmand et al. supported that the prevalence of PE varies considerably in relation to the study setting, ranged from 3.3% to 29.1% in patients with a clinical diagnosis of AE-COPD [36] (Table 2). Bed resting, patient’s comorbidities, and advanced age contribute to that as well [37]. AEs are associated with increased inflammation, hence enhanced thrombotic risk. In particular, according to Aleva et al. [37], PE prevalence was 16.1% (95% CI 8.3–25.8) during an AE of COPD in a total of 880 patients while DVT prevalence was 10.5% (95% CI 4.3–19.0) in a total of 831 patients (Table 2). More recent results by Dentali et al. [38] revealed a PE prevalence of 12.7% (95% CI 10.7–14.8) among 1043 patients hospitalized with AE-COPD and suspected PE, a DVT prevalence of 6.4% (95% CI 5.0–8.1) among 178 patients who were tested with compression ultrasound. The authors reported an isolated DVT prevalence of 2.7% (41.8% of the DVT cases) among the whole study population. However, compression ultrasound was not used to confirm a diagnosis of DVT in all patients but only in 178 patients with signs of DVT [38]. Although DVT is usually twice as frequent as PE in the general population, PE seems to be the most frequent clinical presentation of VTE in patients with AE of COPD [39] (Table 2). Hence, according to the Global Initiative for Chronic Obstructive Lung Disease (GOLD) recommendations [21] and American College of Chest Physicians (ACCP) guidelines [40], thromboprophylaxis should be applied to hospitalized patients with severe COPD exacerbation given the high VTE risk. Nevertheless, VTE events might occur despite thromboprophylaxis during an AE of COPD [34].

Dentali et al. supported that except from the signs and symptoms of DVT (OR: 4.4, 95% CI 2.7–7.4), the presence of a normal chest x-ray (OR: 2.0, 95% CI 1.3–3.1), the partial pressure of carbon dioxide (pCO2) less than 40 mmHg (OR: 1.5, 95% CI 1.0–2.3), hypertension (OR: 1.5, 95% CI 1.0–2.3), and the increasing age (OR: 1.03, 95% CI 1.01–1.06) and the female gender have been associated with higher PE prevalence in COPD patients [38]. Interestingly, the number of the patient’s coexisting risk factors (0, 1, 2, 3, ≥4) was positively correlated with a progressive increase in PE prevalence from 1.8 to 30.4% [38]. This accounts for a decrease of 57 CTPAs needed to diagnose one PE with no risk factors to three CTPAs for a PE diagnosis in a COPD patient with four risk factors. The same study found that PE prevalence was lower in patients with active cancer and patients with purulent sputum [38]. Furthermore, pleuritic chest pain and cardiac failure were more frequent in AE-COPD patients with PE, while respiratory tract infection was less frequently observed in PE patients than non-PE COPD patients [34,37]. Akpinar et al. [43] reported that pleuritic chest pain appears twice more frequently in PE patients. Therefore, identifying the presence and aggregation of the aforementioned parameters may be useful before performing CTPA on patients during an AE of COPD with suspected PE [38].

Regarding ICU patients, the prevalence of PE incidence has been reported to be 13.7%, according to Hassen et al. [30] (Table 2). Increased sputum volume (OR: 0.1, 95% CI 0.0–0.4, *p* = 0.001), recent immobilization for seven days or more (OR: 5.0, 95% CI 1.5–17.2, *p* = 0.01), age 70 years or more (OR: 5.5, 95% CI 1.3–23.7, *p* = 0.02), and invasive mechanical ventilation (OR: 3.6, 95% CI 1.0–13.0, *p* = 0.049) were independently positively associated with concomitant PE events in AE-COPD patients required mechanical ventilation [30]. Nadroparin prophylaxis in mechanically ventilated patients due to acute COPD resulted in a 45% decrease of VTE compared to placebo, with no difference in adverse events [44].

Some challenges might appear while diagnosing PE in COPD patients. Dentali et al. [38] noted that the accuracy of clinical probability scores like Wells and Geneva score needs to be validated in COPD patients. Especially in Wells score, the subjective variable “alternative diagnosis more likely than PE” is quite difficult to be assessed by physicians in AE of COPD where the symptoms of both diseases overlap. D-dimer testing is a tool used to exclude PE, but d-dimers may also increase due to inflammation during acute exacerbations. Gunen et al. [42] reported that 95% of hospitalized patients for AE COPD with VTE had increased d-dimer levels above 500 μg/L. At this cut-off level, the negative predictive value was 0.98. The absence of respiratory infection symptoms and plasma D-dimer elevation above 500 μg/L were significant predictors of PE in patients with COPD exacerbation [45].

As far as CTPA is concerned, it has been supported that in severe AEs of COPD, DVT prevalence is lower than PE, suggesting that many PE cases were in situ thrombotic events rather than embolic events [37]. It is of great clinical significance that only 32.5% of PE events were diagnosed with isolated subsegmental emboli and the rest 68% were diagnosed with a thrombus localized at the segmental or more proximal level (35% in main pulmonary artery, 31.7% in lobar/interlobar arteries and 0.8% in the pulmonary trunk) [37]. Contradictory data are also present reported that emboli tend to lodge in more peripheral sites and are more likely to be unilateral and singular in COPD patients [46].

COPD patients with PE have a worse prognosis, as expected, with increased mortality and length of hospitalization [37]. Hassen et al. [30] reported that mortality was 44% in PE patients than 11% in non-PE COPD patients who admitted to ICU department due to an acute exacerbation. After adjusting for age, a COPD patient with PE had a seven-fold rise in the probability of death (OR: 7.1, 95% CI 0.0–24.9, *p* = 0.002). In addition, the high simplified acute physiology score (SAPS II) (OR: 1.1,95% CI 0.0–1.1, *p* = 0.020) and prolonged mechanical ventilation (OR: 1.1, 95% CI 0.0–1.2, *p* < 0.001) were independent predictors of nosocomial mortality [30]. On the other hand, in many cases, the risk of bleeding outweighs anticoagulation benefits, which placed a question mark in routinely using CTPA for screening. CTPA should rather be used in cases with atypical exacerbation symptoms such as pleuritic pain, signs of cardiac failure, absence of infectious origin, and VTE history [46].

Furthermore, up to 50% of COPD patients may present with pulmonary arterial hypertension (PAH) [47]. PAH is associated with hypercoagulability, leading to in situ thromboses [48]. There have been reported increased levels of von Willebrand factor, TF and PAI-1, and increased shear stress and platelet activation along with the increase of pressure in pulmonary arteries. Considering that PAH is disproportionately severe in 2–5% of all COPD patients [47], the risk of PE in these patients is even greater, as is the prognosis.

Data support that PE during COPD exacerbation is considered provoked, with the most frequent triggering factor being the immobilization due to acute medical illness [39]. Several theories suggest that the risk of VTE recurrence after stopping anticoagulation therapy is similar between COPD and non-COPD patients [49] and the clinical presentation of VTE recurrence to the first VTE event is similar in both groups, while no differences in risk of death after cessation of anticoagulation were found between the two groups. What is less clear is the short- and long-term thrombogenic status that remained after an AE of COPD.

In conclusion, COPD patients carry a high VTE risk that sets them in need of thromboprophylaxis during AE. A higher incidence of PE than DVT suggests that PE cases might occur as in situ thrombosis rather than embolic events. Diagnostic uncertainty could contribute to diagnostic errors. Thus, clinical probability scores and d-dimer testing should be used cautiously due to the overlap of symptoms between PE and COPD.

## 4. Interstitial Lung Diseases and VTE

Interstitial lung diseases (ILDs) are a group of diffuse parenchymal lung disorders resulting in high mortality rates. According to the 2013 classification update by the American Thoracic Society (ATS)/European Respiratory Society (ERS), ILDs are distinguished as major idiopathic interstitial pneumonias (IIPs) (idiopathic pulmonary fibrosis, idiopathic nonspecific interstitial pneumonia, respiratory bronchiolitis-interstitial lung disease, desquamative interstitial pneumonia, cryptogenic organizing pneumonia, acute interstitial pneumonia), rare IIPs (idiopathic lymphoid interstitial pneumonia, idiopathic pleuroparenchymal fibroelastosis) and unclassifiable IIPs [50]. The main representative of this heterogenous group is idiopathic pulmonary fibrosis (IPF), a chronic and progressive disease characterized by extracellular matrix deposition, resulting in lung remodeling. It represents about 20% of ILDs being the most common, carrying the worst prognosis amongst them [51].

An association between IPF and increased risk of thromboembolic events has been documented. The exact cause of the correlation between IPF and VTE is unknown but reduced mobility of patients experiencing symptoms like shortness of breath, fatigue and joint stiffness may predispose to VTE [52]. Additionally, a higher prevalence of pulmonary hypertension (PH) observed in these patients leads to increased resistance of pulmonary arteries and blood stagnation [41], while increased angiogenic chemokines result in aberrant angiogenesis and endothelial abnormalities [53,54,55]. Another explanation might be a common pathogenetic pathway between IPF and VTE. This is supported by the findings of increased thrombin concentration in bronchoalveolar lavage of fibrotic lung disease patients [56], which is considered to be a potent inducer of fibrogenic cytokine generation such as transforming growth factor (TGF)-β, connective tissue growth factor (CTGF) and platelet-derived growth factor-AA [57]. These results are consistent with the study of Imokawa et al. [58] shown increased TF and fibrin depositions on type II pneumonocytes of patients with pulmonary fibrotic disease. Other studies reported decreased protein C activation in IPF patients [59] and induction of TF and PAI-1 [60]. Navaratnam et al. [61] stated that a prothrombotic state in patients with IPF is almost five times higher (OR: 4.8, 95% CI 2.9–7.8) and is associated with increased mortality (HR: 3.5, 95% CI 1.1–9.8).

This procoagulant state elevates VTE risk. Sprunger et al. [62] reported that VTE prevalence in IPF decedents was 1.74% and 34% higher than in the general population (OR: 1.3, 95% CI 1.3–1.4). They also reported that VTE prevalence in IPF decedents was higher than in COPD (OR: 1.4, 95% CI 1.4–1.5) or lung cancer decedents (overall OR: 1.5, 95% CI 1.5–1.6) (Table 3). A previous study by Hubbard et al. supported that the risk of DVT was greater before (OR 2.0, 95% CI 1.1–3.5) and even greater after (rate ratio 3.4, 95% CI 1.6–7.3) the diagnosis of IPF in comparison to the general population [63] (Table 3). According to Dalleywater et al. [64] the prevalence of PE and DVT in IPF patients has been 2.4% (6-fold risk) and 1.1% (2-fold risk), respectively. A meta-analysis by Boonpheng et al. [65] showed that the pooled risk ratio of VTE in IPF was 2.1 (95% CI 1.3–3.5), while Margaritopoulos et al. [52] reviewed a VTE prevalence of 2% in IPF patients, two-fold higher than healthy individuals.

A Danish 27-year study by Sode et al., comprising 7.4 million individuals, estimated an adjusted HR of 2.4 (95% CI 2.3–2.6) and 1.3 (95% CI 1.2–1.4) for PE and DVT, respectively among IPF patients [66]. The same study also revealed an adjusted HR of 2.2 (95% CI 2.1–2.3) and 1.4 (95% CI 1.3–1.4) for PE and DVT in ILD patients (Table 3). Luo et al. [65] reported that among 57 patients with ILDs, 26.3% had VTE [66]. In a retrospective cohort study by Park et al. [41] the prevalence rates for PE and DVT was 1746 (OR: 16.4, 95% CI 9.7–27.4) and 582 (OR: 4.4, 95% CI 1.8–10.6) per 100,000 population, respectively, in ILD patients. The prevalence of PE in ILDs was similar to COPD and higher than the PE prevalence of the general population and patients with connective tissue diseases [41]. DVT prevalence was similar among the patient groups [41]. Interestingly PE/DVT ratio was 3.1 for ILD, 1.8 for COPD, 0.7 for CTD and 0.8 for the general population. This data suggests that there might be in situ thrombosis in pulmonary arteries or that the vascular beds are affected in ILD and COPD individuals, resulting in a delayed or no resolution of the clot, while the process is normally performed in the primary position of DVT [51].

Luo et al. also supported that symptoms like dyspnea (OR: 3.8, 95% CI 1.1–12.8, *p* = 0.035), lower extremity edema (OR: 8.7, 95%CI 1.8–41.4, *p* = 0.007) and palpitations (OR: 4.8, 95% CI 1.1–21.0, *p* = 0.040), as well as d-dimer levels higher than 500 ng/L (OR: 5.1, 95% CI 1.0–25.5, *p* = 0.048) were associated with VTE events in patients with ILDs [67]. Concerning diagnosis, PE should be suspected when there is acute deterioration with simultaneously no imaging features in high resolution computed tomography (HRCT) or findings from serial lung function tests suggesting progression of ILD [52]. Respiratory symptoms due to ILD may interfere with the applicability of clinical probability scores, especially the Wells score, due to the subjective criterion [the physicians’ judgment of whether an alternative diagnosis is less likely than PE]. However, the area under curve (AUC) for the diagnostic performance of Wells and Geneva scores was not significantly different in patients with ILDs (0.720 vs. 0.704) [67]. On the contrary, V/Q scan use should be limited because possible mismatch regions that would otherwise be diagnostic, correspond to honeycombing or emphysema [52].

In the general population, treatment of the acute phase of PE includes parenteral heparin with or without overlap with Vitamin K antagonists (VKA), or alternatively, the use of direct oral anticoagulants DOACs. Nevertheless, in IPF, warfarin has designated cautious use. In a randomized placebo-controlled trial, warfarin was associated with a lack of efficacy and increased mortality [68]. Margaritopoulos et al. [42] suggested that warfarin should be avoided in IPF, noting, however, that there has been no evidence against warfarin use in non-IPF ILDs. The prognosis of IPF patients becomes worse in the presence of VTE. There is no gender-related difference in mortality. Female patients with VTE died earlier (median age: 74.3 vs. 77.4) and the same as male patients with VTE (median age: 72 vs. 74.4) [62].

Sarcoidosis is a systemic inflammatory, granulomatous disease, mainly affecting lung parenchyma. It is associated with increased VTE prevalence as well. According to a meta-analysis by Ungprasert et al. [69] the pooled risk ratio of VTE in sarcoidosis was 1.4 (95% CI 1.1–1.8), while in another study by Ungprasert et al. [69], the HR for VTE, DVT and PE was 3.0 (95% CI 1.5–6.3), 3.1 (95% CI 1.3–7.5), and 4.3 (95% CI 1.2–15.2), respectively. After excluding the first six months following the index date of sarcoidosis diagnosis, to avoid surveillance bias due to more frequent thoracic imaging, the HR for VTE, DVT and PE were 2.7 (95% CI, 1.3–5.7), 3.0 (95% CI 1.3–7.2), and 3.6 (95% CI 1.0–13.0), respectively [70].

To conclude, patients with ILDs or sarcoidosis are at increased risk of VTE. In patients with ILDs and especially IPF, PE prevalence seems to be higher than DVT prevalence, given their potentially damaging effects directly to the lung parenchyma and vessels, leading to in-situ thrombotic events. Further research is needed regarding early diagnosis and treatment safety and efficacy concerning warfarin in patients with ILDs.

## 5. Tuberculosis and VTE

Tuberculosis (TB) is a bacterial disease caused by Mycobacterium Tuberculosis, part of the M. Tuberculosis complex. It is mainly an airborne infectious disease. Any organ can be affected by TB, but in HIV-negative individuals, the pulmonary disease is the most common clinical manifestation (70–80% of cases) [71].

TB is an inflammatory disease associated with activation of the coagulation cascade. More specifically, increased levels of fibrinogen, factor VIII, PAI-1, and decreased antithrombin III and protein C levels have been reported in the plasma of TB patients [71,72]. Most of these parameters went back to normal on the thirtieth day of TB treatment [72,73]. In addition, studies showed that platelets are important players in the formation and function of granuloma and macrophage transformation in TB [74].

Many studies [75,76,77] suggested that the proinflammatory cytokines IL-1, IL-6 and tumor necrosis factor-alpha (TNF-a) are produced by mononuclear or macrophage cells during TB leading to increased thrombotic tendency and up-regulation of coagulation. Moreover, Mycobacterium Tuberculosis itself can induce TF expression through direct contact of bacteria with macrophages. TB is frequently associated with other chronic infections such as HIV infection and viral hepatitis. Hence, these patients may be at increased risk of thrombosis due to the aggregated inflammatory profile in case of co-infection [78]. In the study by Borjas-Howard et al. [78] HIV co-infection was independently associated with VTE (adjusted OR: 8.2, CI 95% 2.9–22.7) (Table 4).

Far too little attention has been paid to the coagulation state of patients with latent tuberculosis infection (LTBI). Shitrit et al. found that patients with LTBI had normal d-dimer levels suggesting that low-level inflammations such as LTBI do not lead to a hypercoagulable state [79]. An increased abundance of vitamin K-dependent protein S (PROS) in both unstimulated and stimulated plasma samples in active TB as compared to LTBI suggests that only patients with TB are in a systemic hypercoagulable state [80].

Additionally, TB infection induces enlargement of lymph nodes that may result in compression of pulmonary veins, which in combination with patients’ bed rest leads to the stagnation of blood [5]. Furthermore, it is suggested that rifampicin, part of TB treatment, increases the risk of DVT (Relative Risk: 4.7, 95% CI 2.9–7.9) compared with other regimens, especially the first two weeks of treatment [81,83].

VTE prevalence is higher in patients with TB. Dentan et al. [81] conducted a large database retrospective study with 27,659,947 TB admissions, and reported a VTE prevalence of 2.1% (95% CI, 1.6–2.6%) in TB patients. Tuberculosis could independently predict VTE with an OR of 1.6 (95% CI 1.2–2.0), close to VTE risk from neoplasia in MEDENOX [84] study (OR: 1.6, 95% CI 0.9–2.8) (Table 4). A meta-analysis revealed that the VTE, PE and DVT prevalence among patients with active TB was 3.5% (95% CI 2.2–5.2) among 16,190 participants, 5.8% (95% CI 2.2–10.7) among 5512 participants and 1.3% (95% CI 0.0–4.1) among 12,928 participants, respectively [5]. The OR for VTE, PE and DVT in these patients were 2.9 (95% CI 2.3–3.7), 3.6 (95% CI 2.5–5.1) and 2.5 (95% CI 1.8–3.4), respectively [5] (Table 4). The increased prevalence of PE compared to DVT is probably affected by the increased frequency of CT performed in TB patients but still seems to be 4–8 times higher than the general population, suggesting that patients with active TB are undoubtedly at increased risk of thrombosis [5].

Borjas-Howard et al. [78] studied the risk of VTE in 750 patients, of which 18 (2.4%, 95% CI 1.4–3.8) suffered a VTE event. The median age at the time of VTE diagnosis was 42 years and 13 out of 18 VTE events happened in the time window of two weeks before the anti-TB treatment initiation.

The use of VKA in TB patients experiencing a thromboembolic event can be complicated by drug interactions and especially by rifampicin, which is used as a first-line drug for TB. Warfarin is a racemic mixture of both R and S enantiomeres. The R enantiomer is less potent than the S, and has a longer half-life. R warfarin is metabolized by the enzymes cytochrome P450 (CYP450), in particular, CYP 1A2 and CYP 3A4, while S warfarin is metabolized by CYP 2C9 [85]. Rifampicin is a potent and non-specific inducer of the CYP450 enzyme system, and especially a recognized inducer of CYP 3A4, enhancing the metabolism of both enantiomeres. Rifampicin accelerates warfarin’s clearance, resulting in decreased anticoagulant effect and difficulty in achieving the therapeutic INR level. In a study by Maina et al. [85] of 10 patients receiving both rifampicin and warfarin, the median TTR (Time in Therapeutic Range) was 47% compared to the TTR for those not treated with rifampicin which was 62%. In another study, the median TTR was only 33% [86]. In such cases, abruptly discontinuing rifampicin before the completion of TB treatment is potentially hazardous. Failing achieving the appropriate TTR may result in residual clots and potentially in chronic thromboembolic pulmonary hypertension. DOACs could be an alternative for anticoagulation. However, data on their efficacy and safety in such cases are only a few [87].

Mortality of concomitant active TB and VTE is reported to be about 15%, while mortality from isolated TB or VTE is 2.7% or 2.5%, respectively [81]. VTE was independently associated with mortality (OR: 3.9, 95% CI 1.9–8.1) in TB patients.

To sum, TB is a prothrombotic disease. HIV co-infection and even TB treatment must raise awareness in managing these patients. Interaction between rifampicin and warfarin amplifies the need for further research in the use of DOACs in TB cases.

## 6. In Situ Thrombosis of the Pulmonary Arteries: Another Piece to the Puzzle of Acute VTE in Patients with Chronic Inflammatory Lung Diseases?

Table 1 summarizes the PE, DVT, and total VTE prevalence in the most common chronic airway and parenchymal inflammatory lung diseases. All of the studies reviewed here support the hypothesis that patients with chronic inflammatory lung diseases are at increased risk of a first episode of PE as well as recurrent episodes of PE rather than DVT, and greater risk of mortality [88].

The increased frequency of chest imaging in patients with pulmonary diseases may lead to detection biases causing associations to be larger for PE than DVT. However, PE without leg thrombosis has been reported in up to 50% of PE patients [10]. A pelvic or abdominal thrombosis using magnetic resonance angiography was detected only in 29% of patients with acute PE [89,90]. No peripheral clot was found in 56% of patients with a first PE episode, using magnetic resonance direct thrombus imaging, a method with high sensitivity and specificity of 98% and 96%, respectively, that detects a clot for up to 6 months based on the transformation of hemoglobin into methemoglobin [90,91]. The authors in that study hypothesized that the clot could be of cardiac origin. However, no patient had signs consistent with heart failure (upon physical examination) and only two out of 55 patients with PE and no DVT had atrial fibrillation, making this assumption unlikely. Another possible explanation could be that the whole clot was completely dislodged from its primary position and ended up in pulmonary arteries. This theory seems unlikely for all the missing cases of DVT as it is not consistent with the data of post-mortem studies.

Hence, an occlusion of pulmonary arteries is not always due to embolism but might be an in-situ thrombosis, as here suggested. The same conditions that lead to the formation of the clot in peripheral sites might also lead to the primary clot formation in pulmonary arteries in cases of a chronic inflammatory lung disease.

## 7. Conclusions

The presence of chronic inflammatory lung diseases, such as asthma, COPD, ILDs, or active TB, increases the risk of VTE, especially PE, compared to the general population. High suspicion is needed to diagnose a thrombotic event early as the clinical presentation inevitably overlaps with respiratory disorders. PE risk increases with disease severity. The elevated inflammatory burden during disease exacerbations seems to predispose to VTE events correlated with increased mortality risk. Hospitalized patients should be considered for thromboprophylaxis administration, given the presence of additional risk factors during hospitalization such as immobilization, respiratory failure, and use of central venous catheters. The corticosteroid administration in asthma and COPD independently increases the risk of VTE, while TB treatment with rifampicin requires anticoagulant treatment with warfarin. This combination of findings provides some support for the conceptual premise that connections exist between VTE and chronic lung inflammatory comorbidities. Hence, it could conceivably be hypothesized that VTE patients should be asked for lung comorbidities before determining the duration of anticoagulant therapy. However, further research should be undertaken to understand the underlying pathophysiology of in situ or de novo thrombosis in those patients.

## Figures and Tables

**Table 1 jcm-10-02061-t001:** Characteristics of studies included in this review regarding the relationship between asthma and venous thromboembolic disease (VTE).

Study	Type of Study	Timeframe	Population	VTE Incidence Rate	Standardized Rate Ratio for VTE	Adjusted Odds Ratio for VTE	Adjusted Hazard Ratio for VTE	Estimated Time of Risk VTE
Zöller et al., 2017 [24]	Nationwide case-control retrospective study	1981–2010	114,366 Swedish-born asthma patients with hospital PE diagnosis, 76,494 asthma patients with DVT, 6854 asthma patients with both PE and DVT Age-, gender-and eductional attainment matched controls for each case	Not provided	Not provided	PE: 1.4 (95% CI 1.4–1.5) *	Not provided	1981–2010
						DVT: 1.6 (95% CI 1.5–1.7) *		
						Combined PE/DVT: 1.6 (95% CI 1.3–2.0) * * adjusted for comorbidities		
Chung et al., 2014 [17]	Nationwide-population retrospective study	2002–2008	31,356 patients with newly diagnosed asthma, 125,157 age- and gender-matched nonasthmatics	Not provided	PE: 3.3 (95% CI 3.2–3.4)	Not provided	PE: 3.2 (95% CI 1.7–6.0) *	Until the end of 2010
							PE: 3.4 (95% CI 1.7–6.7) * * adjusted for sex, age, comorbidities and estrogen use	During the first 5 years after asthma diagnosis
Majoor et al., 2013 [23]	Prospective study	1 December 2010–1 May 2011	648 asthma patients (283 severe and 365 mild-to-moderate asthma patients) visiting outpatient asthma clinics	PE: Severe asthma: 0.9 (95% CI 0.4–1.4) per 1000 person-years Mild-to-moderate asthma: 0.3 (95% CI 0.1–0.6) per 1000 person-years General population: 0.2 (95% CI 0.0–0.3) per 1000 person-years	PE: Severe asthma: 8.9 (95% CI 4.6–15.6) Mild to moderate asthma: 4.0 (95% CI 1.0–9.1)	Not provided	PE: Severe asthma: 3.3 (95% CI 1.2–9.9)	Mean risk time: 39 years (range 20–63 years)
				DVT: Severe asthma: 0.4 (95% CI 0.0–0.7) per 1000 person-years, Mild to moderate asthma: 0.6 (95% CI 0.3–1.0) per 1000 person-years General population: 0.3 (95% CI 0.1–0.5) per 1000 person-years	DVT: Severe asthma; 1.6 (95% CI 0.4–4.1) Mild to moderate asthma: 1.5 (95% CI 0.3–4.2)		DVT: was not associated with asthma	

Abbreviations: CI, confidence intervals; DVT, deep vein thrombosis; PE, pulmonary disease; VTE, venous thromboembolic disease.

**Table 2 jcm-10-02061-t002:** Characteristics of studies included in this review regarding the relationship between chronic obstructive pulmonary disease (COPD) and venous thromboembolic disease (VTE).

Study	Type of Study	Timeframe	Population	Prevalence of VTE	Relative Risk	Estimated Time of Risk
Dentali et al., 2020 [38]	Multicenter retrospective cohort study	1 January 2011–31 December 2011	1043 patients with an AE-COPD and suspected PE	PE: 12.7% (95% CI 10.7–14.8)	Not provided	Not provided
				DVT: 6.4% (95% CI 5.0–8.1) (among 178 patients tested)	Not provided	
Hassen et al., 2020 [30]	Prospective cohort study	March 2013–May 2017	131 patients with AE-COPD requiring mechanical ventilation	PE: 13.7%	Not provided	Not provided
Pourmand et al., 2018 [36]	Systematic review	1990–2017	Sample sizes of 5 studies ranged from 49–197 patients admitted to the hospital with a clinical diagnosis of AE-COPD	PE: 3.3–29.1% according to the study setting	Not provided	Not provided
Aleva et al., 2017 [37]	Systematic review and meta-analysis	1974–October 2015	880 patients with AE-COPD admitted or hospitalized (7 studies)	PE: 16.1% (95% CI 8.3–25.8)	Not provided	Not provided
				DVT: 10.5% (95% CI 4.3–19.0) (among 831 patients)	Not provided	
Park et al., 2016 [41]	Retrospective study	January 2011–December 2011	15,686 COPD patients 859 ILD patients 640,177 general population and 7280 CTD patients	PE: 11.3 (95% CI 9.6–13.2)	Not provided	One year risk
				DVT: 4.8 (95% CI 3.9–5.9)		
Gunen et al., 2010 [42]	Prospective cohort study		131 patients with AE-COPD	PE: 13.7% (95% CI 7.8–19.6)	2.528 (95% CI 1.144–5.588)	One year risk
				DVT: 10.6% (95% CI 5.3–15.9)		
Rizkallah et al., 2009 [35]	Systematic review and meta-analysis	1949–April 2008	550 patients who did and did not require hospitalization (5 studies)	PE: 19.9% (95% CI 6.7–33.0) of the whole study group	Not provided	Not provided
				PE: 24.7% (95% CI 17.9–31.4) in hospitalized patients	Not provided	

Abbreviations: AE-COPD, acute exacerbation of chronic obstructive pulmonary disease; CTD, connective tissue disease; CI, confidence intervals; DVT, deep vein thrombosis; ILD, idiopathic pulmonary disease; PE, pulmonary disease; VTE, venous thromboembolic disease.

**Table 3 jcm-10-02061-t003:** Characteristics of studies included in this review regarding the relationship between interstitial lung diseases (ILDs) and venous thromboembolic disease (VTE).

Study	Type of Study	Timeframe	Population	Prevalence of VTE	Adjusted Odds Ratio for VTE	Rate Ratio for VTE	Risk Ratio for VTE	Adjusted Hazard Ratio for VTE	Estimated Time of Risk
Boonpheng et al., 2018 [65]	Systematic review and meta-analysis	February 2017–2018	Sample sizes of 5 studies ranged from 211–218,991 IPF patients	Not provided	Not provided	Not provided	Pooled risk ratio: 2.1 (95% CI, 1.3–3.5)	Not provided	Not provided
Park et al., 2016 [41]	Retrospective cohort study	January 2011–December 2011	859 ILD patients 640,177 general population 15,686 COPD and 7280 CTD patients	Not provided	PE: 16.4 (95% CI 9.7–27.4)	Not provided	Not provided	Not provided	One year risk
					DVT: 4.4 (95% CI 1.8–10.6)				
Dalleywater et al., 2014 [64]	Prospective database cohort study	2000–2006	3211 incident cases of IPF-clinical syndrome and 12,307 matched controls	PE: 2.4%	Not provided	PE: 9.3 (95% CI, 7.4–11.7) *	VTE: 3.7 (95% CI, 1.2–11.0)	Not provided	Median (interquartile range) follow-up after the index date was 1.7 (0.6–3.6) years in cases and 3.3 (1.5–5.8) years for controls.
				DVT: 1.1%		DVT: 4.3 (95%CI 3.0–6.0) * * adjusted for matching variables, smoking habit and warfarin prescription		Not provided	
Sprunger et al., 2012 [62]	Cross-sectional study database study	1988–2007	218,991 records with IPF	VTE: 1.74%	VTE: 1.3 (95% CI 1.3–1.4) * * adjusted for age, sex and year of death	Not provided	VTE: 1.3 (95% CI, 1.3–1.4)	Not provided	1988–1998
Sode et al., 2009 [66]	Retrospective cohort	1980–2007	19,557 individuals with IPF and 34,493 individuals with ILDs and 7260,277 controls	Not provided	Not provided	Not provided	Not provided	IPF: PE 2.4 (95% CI, 2.3–2.6)	Not provided
								IPF: DVT: 1.3 (95% CI, 1.2–1.4)	
								ILD: PE: 2.2 (95% CI, 2.1–2.3)	
								ILD: DVT: 1.4 (95% CI, 1.3–1.4)	
Hubbard et al., 2008 [63]	Comparative retrospective and prospective database study	1991–2003	920 incident case subjects of IPF and 3593 matched control subjects	Not provided	DVT: 2.0 (95% CI, 1.1–3.5) * * adjusted for age, sex, smoking habit and medications	Not provided	VTE: 3.4 (95% CI, 1.6–7.3)	Not provided	Mean duration before index date: 7.8 ± 3.9 years for case subjects (7.7 ± 3.9 years for control subjects)

Abbreviations: CI, confidence intervals; DVT, deep vein thrombosis; ILD, interstitial lung disease; IPF, idiopathic pulmonary fibrosis; PE, pulmonary disease, OR, odds ratio; VTE, venous thromboembolic disease.

**Table 4 jcm-10-02061-t004:** Characteristics of studies included in this review regarding the relationship between tuberculosis and venous thromboembolic disease (VTE).

Study	Type of Study	Timeframe	Population	Prevalence of VTE	Odds Ratio for VTE	Estimated Time of Risk
Danwang et al., 2021 [5]	Systematic review and meta-analysis	Until 15 December 2019	16,190 participants (9 studies included)	VTE: 3.5% (95% CI 2.2–5.2) 16,190 participants (9 studies included)	VTE: 2.9 (95% CI 2.3–3.7)	Not provided
				PE: 5.8% (95%) CI 2.2–10.7) 6 studies; 5512 participants	PE: 3.58 (95% CI 2.5–5.1)	
				DVT: 1.3% (95% CI 0.0–4.1) 5 studies; 12,928 participants	DVT: 2.47 (95% CI 1.8–3.4)	
Borjas-Howard et al., 2017 [78]	Retrospective study	2000–2010	750 participants	2.4 (95% CI 1.4–3.8)	VTE in HIV-positive: 8.2 (CI 95% 2.9–22.7) * * Adjusted for gender and hospitalization	Not provided
Dentan et al., 2014 [81]	Retrospective database cohort study	1 January 2006–31 December 2006	27,659, 947 TB admissions	VTE: 2.1% (95% CI, 1.6–2.6)	VTE: 1.55 (95% CI, 1.2–2.0)	One year risk
				PE: 0.9% (95% CI, 0.7–1.3)		
				DVT: 1.1% (95% CI 0.8–1.5)		
Marjani et al., 2012 [82]	Prospective study	2007–2009	1153 participants	VTE: 2.8% (95% CI, 1.9–3.9)	Not provided	Not provided
				PE: 1% (95% CI, 0.5–1.7)		
				DVT: 2% (95% CI, 1.3–3)		

Abbreviations: CI, confidence intervals; DVT, deep vein thrombosis; PE, pulmonary disease, OR, odds ratio; VTE, venous thromboembolic disease.

## Data Availability

All data generated or analysed during this study are included in this published article.

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
