# Peer review of "Venous Thromboembolic Disease in Chronic Inflammatory Lung Diseases: Knowns and Unknowns"

_jcm, 2021, doi:10.3390/jcm10102061_

Round 1
Reviewer 1 Report
The Authors present a narrative review providing an overview of the evidence on the possible association between chronic inflammatory lung diseases and venous thromboembolism (particularly pulmonary embolism). The topic is certainly of interest; the review shows that a good effort was made to collect an adequate amount of evidence; and the quality of written English is relatively high.
Unfortunately, however, I feel that the article is not ready for publication in its present form, for three problems – two major and one minor. I detail the three problems below and provide then suggestions for the Authors on how to re-think the article. .
PROBLEMS
1) The Authors conduct a narrative review. However, the topic is already addressed by systematic reviews or even meta-analyses.
Specific examples below.
COPD and Pulmonary embolism, there are at least three relatively recent systematic reviews/meta-analyses:
Pourmand A, Robinson H, Mazer-Amirshahi M, Pines JM. Pulmonary Embolism Among Patients With Acute Exacerbation Of Chronic Obstructive Pulmonary Disease: Implications For Emergency Medicine. J Emerg Med. 2018 Sep;55(3):339-346. doi: 10.1016/j.jemermed.2018.05.026. Epub 2018 Jun 23.
Aleva FE, Voets LWLM, Simons SO, de Mast Q, van der Ven AJAM, Heijdra YF. Prevalence and Localization of Pulmonary Embolism in Unexplained Acute Exacerbations of COPD: A Systematic Review and Meta-analysis. Chest. 2017 Mar;151(3):544-554. doi: 10.1016/j.chest.2016.07.034. Epub 2016 Aug 12. PMID: 27522956
Rizkallah J, Man SFP, Sin DD. Prevalence of pulmonary embolism in acute exacerbations of COPD: a systematic review and metaanalysis. Chest. 2009 Mar;135(3):786-793. doi: 10.1378/chest.08-1516. Epub 2008 Sep 23. PMID: 18812453
Because COPD is currently considered to include asthma in respiratory medicine, these reviews partly cover asthma as well.
The association with ILD is more original, but there is one relatively recent systematic review and meta-analysis:
Boonpheng B, Ungprasert P. Risk of venous thromboembolism in patients with idiopathic pulmonary fibrosis: a systematic review and meta-analysis. Sarcoidosis Vasc Diffuse Lung Dis. 2018;35(2):109-114. doi: 10.36141/svdld.v35i2.6213. Epub 2018 Apr 28
And one on TBC-VTE is just out:
Danwang C, Bigna JJ, Awana AP, Nzalie RN, Robert A. Global epidemiology of venous thromboembolism in people with active tuberculosis: a systematic review and meta-analysis. J Thromb Thrombolysis. 2021 Feb;51(2):502-512. doi: 10.1007/s11239-020-02211-7.
2) The Authors report effect sizes narratively. However, these are difficult to interpret without a consistent specification of the population from which they were referred, confidence intervals, and timeframe they refer to. Incidence rates should be distinguished from cumulative incidence rates and the measurement unit should always be reported. Effect sizes should be rounded consistently to one or, only if necessary, two significant digits.
Specific examples below.
- The Authors report several HRs and OR in the text. However, loose effect sizes are not informative at all. In general, measures of disease frequency and measures of association (risk) require this information: 1) in which population, 2) which effect size, 3) a measure of uncertainty (usually a confidence interval), and 3) the timeframe to which the effect size refers. Isolated HRs require 1) a confidence interval; and 2) the information whether they were univariate or adjusted and, if they were adjusted, for what. For ORs/RRs and other binomial outcome measures, a third additional and very important concern exists: 3) in what timeframe? For instance, the authors mention under “Asthma and VTE”: “it was reported that the odds ratio was 1.43 for PE, 1.57 for DVT, and 1.62 for both entities in asthma patients compared to healthy individuals”. But is this the lifetime risk, the 10-year risk, the 5-year risk?
- Another example of this uncertainty in the use of statistics. Later on, in “Asthma and VTE: “a nationwide cohort study by Chung et al. showed that PE was three times higher in asthmatic patients compared to non-asthmatic individuals”. It does not make sense to write that “a disease is X times higher”. The author mean some basic epidemiologic measure of disease occurrence: maybe “a history of” (if this was a cross-sectional or case-control nationwide study), or rather (as it was a cohort study, but could it have been nationwide?) “the incidence of”? And if it is “incidence”, the question becomes: in how long? Lifetime risk?
- A subsequent example is a HR that is correctly accompanied by a timeframe: “The HR for PE is 3.38 during the first five years after diagnosing asthma”. However, in this case I question the generalizability imposed by the present tense: we miss the population in which this was tested. If this was a high-quality meta-analysis, I would accept the generalization, but then it should specified whethe rit was.
- Further subtle confusion in technical language: “According to the study of Alzghoul et al. [19], PE incidence was 19.1% in patients 105 during an asthma exacerbation.” I do not think that an INCIDENCE could be calculated during an asthma exacerbation and given as a percentage. I suppose this was a prevalence or, at most, a cumulative incidence; maybe the in-hospital cumulative incidence rate? Or the 30-day cumulative incidence rate?
3) A relatively minor but non-negligible problem: while the quality of written English is generally high, there are several uncertainties in word choice and the use of prepositions. This is widespread throughout the article.
Below, I provide some specific examples from throughout the article that are not at all exhaustive.
- Abstract: “Provide all the available evidence”: the sentence is not wrong, but usually “provide evidence” means directly collect data and make calculations, and is therefore used for original research. Because this is a review, the expression “collect evidence” would be better. Or: “provide an overview of”.
- Abstract: “The relationship between the most common chronic inflammatory lung idseases with VTE”. “Between” is followed by “and”, not “with”. Therefore, the Authors should either write “relationship between A and B” or “relationship of A with B”.
- Introduction: “Many studies support that...” --> Many studies support the notion/idea that.
- Introduction: “no risk factor can be identified at a third to half of VTE events” --> IN a third to half of VTE events.
- Introduction: “predispose a VTE episode” --> “predispose TO a VTE episodes.
- Asthma and VTE: “Asthma has 69 been associated with enhanced procoagulant and antifibrinolytic activities in the airways 70 [9-13], which are activated through TF overexpression mechanisms”: “which are activated” does not seem correct; if it refers to “activities” we have the awkward “procoagulant and antifibrinolytic activities are activated through...” and it should prorably be simply “which, in turn, stem from” or “originate from” or “derive from”. If it refers to “airways”, I am unsure whether it is correct and the Authors should rephrase.
- Asthma and VTE: “a increased risk in both PE and DVT in asthma patient”: this use of “in” makes the phrase unclear. “Risk in” always means: in a given population; “in” never introduces the condition the risk of which or for which increases. Therefore, it should be “an increased risk of both PE and DVT in asthma patient”.
- Some additional inconsistency in the reporting of effect sizes is the number of significant digits: there is no reason to report OR with three significant digits. Page 4, lines 180-183: “Regarding ICU patients, PE incidence is 13.7%, according to Hassen et al. [21]. In-180 creased sputum volume (OR: 0.106), recent immobilization for seven days or more (OR: 181 5.024), age 70 years or more (OR: 5.483) and invasive mechanical ventilation (OR: 3.615) 182 were independently positively associated with PE events [21].
- Various other minor problems in word choice: Page 7, lines 339: “Shitrit et al, found “: it seems that a comma was used instead of a dot; or, alternatively, that a comma was put between the subject and the verb? Page 8, line 370: “In other words” is quite colloquial and not found in scientific writing. Page 8, line 407: “no patient had heart failure findings “: it is quite clear what the author mean, but the rigorous way to write this in English is “no patient had signs consistent with heart failure [upon physical examination]”; “heart failure findings” is an unusual wording. Page 9, line 426: “while TB treatment with rifampicin implicates anticoagulant treatment with warfarin.” What does “implicates” mean here? Does it mean “require”?
SOLUTIONS
I am sympathetic to the Authors’ effort. In order to strengthen their article, I would encourage to address all three problems.
Problem 1) would be addressed by making this a systematic review or even a meta-analysis (although this may require some thought on the comparability of the diseases involved; ILDs have a much lower inflammatory component than the other ones; asthma and COPD are considered within the same spectrum, and they may be considered together with a pre-defined subgroup analysis).
Problem 2) would be addressed by making much greater use of summary tables; this is what most narrative reviews do. Summary tables would allow to always report confidence intervals and measurement units. It would be also easy to separate studies by design (cross-sectional or case-control vs cohort) or reference population (general population, outpatients, non-ICU hospitalized, ICU...). The text can then loosely refer to the tables with overarching statements such as “the in-hospital risk of PE in patients hospitalized for COPD exacerbation was consistently two to three times higher than in patients hospitalized for other reasons in three cohort studies and one case-control studies published between 2000 and 2015 (Table 1)”. Note how this sentence provides detailed information on the comparison group/reference population and the time frame (in-hospital) and for details refers the reader to a table which will include confidence intervals and the specific numbers.
Problem 3) would be solved by a careful revision of all sentences; professional proofreading would of course solve everything, but it may be exaggerated, as the English is not that bad – a native English speaker (even without a medical background) would be more than enough to identify and solve most problems.
I hope the Authors can find a solution and wish them good luck.
Reviewer 2 Report
The review presents the incidence of thrombotic events in respiratory patients and study the possible relationship of underlying inflammation in this patients as a risk factor for thrombosis. They review incidence of thrombosis in asthma, COPD, ILD and TBC but no other infectious diseases.
There is a lack of data on the pathophysiological mechanisms that justify the development of thrombotic events in respiratory diseases.
The authors mention the concept of immunothrombosis but do not develop it.
The conclusions suggest that respiratory diseases should be considered a risk factor for VTE and should be taken into account to deciding the duration of anticoagulant treatment. There is no enough evidence to this asseveration.Author Response
Please see the attachment

Round 2
Reviewer 1 Report
The Authors have made a considerable effort to address the limitations mentioned in the previous review round. In particular, I appreciate these two points:
1) the great improvement in the readability of the effect sizes in the text. Now, the text provides a description of the finding of the individual studies sufficient for the reader to obtain an idea of the corresponding effect sizes and level of evidence, and the reader interested in more detail can consults the tables. These provide now all detail that is needed and are much better than the former, excessively concise Table 1. Providing 4 detailed tables is a good choice; as each table focuses on each of the four diseases treated in the text, in the same order, it is now much easier for the reader to retrieve more detailed information on each of the studies mentioned in the text.
2) In addition, the bibliography is now more representative of the state of the art on the topic, as it includes meta-analyses previously not mentioned.
I understand that a meta-analysis would be beyond the scope of this analysis and recognize the merit of a narrative review covering the scope of all respiratory diseases.
I have only a number of specific points of content (and some additional minor formal notes) that still need addressing after the first review solved most formal issues.
MAJOR/CONTENT-RELATED ISSUES
1) Introduction. I think that discussing concisely the possible pathophysiology underlying most associations between respiratory disease and VTE in the Introduction rather than in the Discussion at the end of the article is a good choice, because pathophysiology is not the core of this article, but the reader needs a quick overview of why and how in general respiratory disease may lead to VTE before the strength of the association of specific diseases with VTE is addressed. The Authors correctly mention immunothrombosis as a general mechanism (or, in truth, a family of mechanisms) linking inflammation and thrombosis, which can be mentioned now because it is common to all respiratory diseases, while mechanisms specific to asthma, COPD, and IPF are correctly discussed further in the respective paragraphs. However, their overview on immunothrombosis and NET-mediated thrombosis is most often used in the field of the association of sepsis and thrombosis, rather than. In a review that specifically focuses on lung diseases (rather than systemic sepsis or pneumosepsis), one should at least discuss the alternative mechanism of non-systemic, but rather local thrombophlebitis mediated by viral or bacterial pulmonary infections. After the words “triggering immunothrombosis”, they may add one sentence mentioning the hypothesis of thrombophlebitis caused directly by viruses or bacteria (rather than mediated by systemic inflammation in any sepsis) with these references:
“Through largely overlapping pathways, and with the possible additional mediation of agent-specific virulence factors or autoantibodies, infection by both viruses (doi: 10.1007/s00277-011-1334-9) and bacteria (doi: 10.1055/a-1177-5127), to which patients with chronic and acute respiratory disease are particularly vulnerable, has been postulated to cause local or systemic thrombophlebitis”.
2) Page 3, lines 110 to 114:
“In addition, Majoor et al. [14] re-110 ported a nine‐fold or a 3.5‐fold increased risk of PE in patients with severe asthma or 111 mild‐moderate asthma compared to the general population, respectively. Interestingly, 112 the use of systemic corticosteroids seemed to increase the risk of VTE, as stated by Jo-113 hannesdottir et al. [17] and Stuijver et al. [18].”
This raises immediately the important question of possible confounding by indication. Did the studies by Hannesdottir et al. en Stuijver et al. exclude confounding by indication? That is, the possibility that the use of systemic corticosteroids was not the cause fo the higher risk of PE, but rather identified patients with higher asthma severity (and it was the latter, not the use of corticosteroids in itself, that led to higher impairment of coagulation and thus to higher risk of VTE)? If these authors admitted that this was a limitation or did not take any measure to control for it (such as adjustment/stratification for disease severity), please add a note such as “However, it cannot be excluded that these findings may have been explained to some extent by confounding by indication, with the use of corticosteroids indicating higher disease severity”. In contrast, if the Authors did take some measure to exclude confounding by indication, mention this – also briefly, for example “Because these effects were adjusted for (...), these results are unlikely to be explained by confounding by indication”.
3) Page 3, lines 99-103: “Contradictory data are also presented showing that the OR 99 declined with the increase of age [15]. In particular, it has been found that the OR for PE 100 and DVT decreased from 2.218 (95% CI 1.9–2.6) and 2.01 (95% CI 1.7–2.4) for patients 101 younger than 50 years of age to 1.54 (95% CI 1.4–1.6) and 1.655 (95% 1.4–1.7) for patients 102 older than 75, respectively [15].”.
This is also intriguing, especially considering that age is the strongest independent risk factor for VTE in the general population (with an exponential increase!). Could the Authors comment in an additional short sentence on the contrast between the finding of reference 8 and reference 15? Is it possible that another form of confounding was present in this case, such as detection bias (older patients receiving less systematic screening, e.g. chest angio CT, than younger patients, in study 15? Conversely, it is possible that reference 8, rather than reference 15, had a worse study design, and that study 15 is correct, because increasing age in those with no asthma compensated for the increase in risk imposed by asthma, so that the difference in risk between asthma and no asthma may have become less evident in older age groups.
4) I suggest that the Authors move the paragraph on the association between asthma and coagulation abnormalities after all paragraphs on the association between asthma and clinical endpoints. That is, the lines 104 to 110 (from “Sneeboer et al.” to “but not with eosinophilic counts”) should be moved after line 136 and before the concluding paragraph (lines 137-144). In this way, the nice paragraph between 137-144 sums up the evidence just presented in the same order (as the “pathophysiological mechanisms” are presented last in the sentence 140-142).
MINOR/FORMAL ISSUES
Would it be not possible to have the tables in the main text rather than as supplementary tables? It is normal for a large review to have several tables. I cannot find a limit to the number of tables in the main manuscript for Journal of Clinical Medicine. Of course, if the Editorial Office sets forth that they should be in the Supplement for space reasons, then keep them in the Supplement.
Page 4, line 137> “In conclusion” may replace “To sum”.
Page 8, line 404: “Studies that were conducted pursuing VTE prevalence showed that patients with TB 404 are at increased risk.”. It would be simpler as “VTE prevalence is higher in patients with TB.”.
Page 9, lines 446-447: The question that forms the title of section 6 is nice, but a little too long. Please shorten it by saving a couple words: “6. In situ thrombosis of the pulmonary arteries: another piece to the puzzle of acute VTE in patients with chronic inflammatory lung diseases?”
Page 9 to 14, lines 456-457: “In pulmonary thrombosis, the 456 embolization of a peripheral clot to the pulmonary arteries might not always be the case”. Maybe clearer if re-formulated as follows: “Pulmonary thrombosis may not always result from embolization of a peripheral clot to the pulmonary arteries”.
Author Response
COMMENTS FROM REVIEWER 1:
- The Authors have made a considerable effort to address the limitations mentioned in the previous review round. In particular, I appreciate these two points: 1) the great improvement in the readability of the effect sizes in the text. Now, the text provides a description of the finding of the individual studies sufficient for the reader to obtain an idea of the corresponding effect sizes and level of evidence, and the reader interested in more detail can consults the tables. These provide now all detail that is needed and are much better than the former, excessively concise Table 1. Providing 4 detailed tables is a good choice; as each table focuses on each of the four diseases treated in the text, in the same order, it is now much easier for the reader to retrieve more detailed information on each of the studies mentioned in the text. 2) In addition, the bibliography is now more representative of the state of the art on the topic, as it includes meta-analyses previously not mentioned. I understand that a meta-analysis would be beyond the scope of this analysis and recognize the merit of a narrative review covering the scope of all respiratory diseases.
RESPONSE: We sincerely thank you for your kind words about our paper. We are delighted to receive a positive feedback from you.
- I have only a number of specific points of content (and some additional minor formal notes) that still need addressing after the first review solved most formal issues.
RESPONSE: We appreciate you taking the time to offer us your comments and insights related to the manuscript. In the following sections, you will find our responses to each of your points and suggestions.
- MAJOR/CONTENT-RELATED ISSUES 1) Introduction. I think that discussing concisely the possible pathophysiology underlying most associations between respiratory disease and VTE in the Introduction rather than in the Discussion at the end of the article is a good choice, because pathophysiology is not the core of this article, but the reader needs a quick overview of why and how in general respiratory disease may lead to VTE before the strength of the association of specific diseases with VTE is addressed.
RESPONSE: Thank you for this great suggestion, that helps to improve the quality of the manuscript. In the revision, as suggested, we discuss the possible pathophysiology underlying most associations between respiratory disease and VTE in the Introduction Section rather than in the Discussion (page 2, lines 47-82).
- The Authors correctly mention immunothrombosis as a general mechanism (or, in truth, a family of mechanisms) linking inflammation and thrombosis, which can be mentioned now because it is common to all respiratory diseases, while mechanisms specific to asthma, COPD, and IPF are correctly discussed further in the respective paragraphs. However, their overview on immunothrombosis and NET-mediated thrombosis is most often used in the field of the association of sepsis and thrombosis, rather than. In a review that specifically focuses on lung diseases (rather than systemic sepsis or pneumosepsis), one should at least discuss the alternative mechanism of non-systemic, but rather local thrombophlebitis mediated by viral or bacterial pulmonary infections. After the words “triggering immunothrombosis”, they may add one sentence mentioning the hypothesis of thrombophlebitis caused directly by viruses or bacteria (rather than mediated by systemic inflammation in any sepsis) with these references: “Through largely overlapping pathways, and with the possible additional mediation of agent-specific virulence factors or autoantibodies, infection by both viruses (doi: 10.1007/s00277-011-1334-9) and bacteria (doi: 10.1055/a-1177-5127), to which patients with chronic and acute respiratory disease are particularly vulnerable, has been postulated to cause local or systemic thrombophlebitis”.
RESPONSE: Thank you for this valuable comment and guidance. In the revised manuscript we added this sentence, supported by the appropriate references (page 2, lines 79-82).
- 2) Page 3, lines 110 to 114: “In addition, Majoor et al. [14] reported a nine‐fold or a 3.5‐fold increased risk of PE in patients with severe asthma or mild‐moderate asthma compared to the general population, respectively. Interestingly, the use of systemic corticosteroids seemed to increase the risk of VTE, as stated by Johannesdottir et al. [17] and Stuijver et al. [18].” This raises immediately the important question of possible confounding by indication. Did the studies by Hannesdottir et al. en Stuijver et al. exclude confounding by indication? That is, the possibility that the use of systemic corticosteroids was not the cause for the higher risk of PE, but rather identified patients with higher asthma severity (and it was the latter, not the use of corticosteroids in itself, that led to higher impairment of coagulation and thus to higher risk of VTE)? If these authors admitted that this was a limitation or did not take any measure to control for it (such as adjustment/stratification for disease severity), please add a note such as “However, it cannot be excluded that these findings may have been explained to some extent by confounding by indication, with the use of corticosteroids indicating higher disease severity”. In contrast, if the Authors did take some measure to exclude confounding by indication, mention this – also briefly, for example “Because these effects were adjusted for (...), these results are unlikely to be explained by confounding by indication”.
RESPONSE: Thank you for this great remark. In the revised manuscript we define this issue on pages 3-4, lines 147-153.
- 3) Page 3, lines 99-103: “Contradictory data are also presented showing that the OR declined with the increase of age [15]. In particular, it has been found that the OR for PE and DVT decreased from 2.218 (95% CI 1.9–2.6) and 2.01 (95% CI 1.7–2.4) for patients younger than 50 years of age to 1.54 (95% CI 1.4–1.6) and 1.655 (95% 1.4–1.7) for patients older than 75, respectively [15].” This is also intriguing, especially considering that age is the strongest independent risk factor for VTE in the general population (with an exponential increase!). Could the Authors comment in an additional short sentence on the contrast between the finding of reference 8 and reference 15? Is it possible that another form of confounding was present in this case, such as detection bias (older patients receiving less systematic screening, e.g. chest angio CT, than younger patients, in study 15? Conversely, it is possible that reference 8, rather than reference 15, had a worse study design, and that study 15 is correct, because increasing age in those with no asthma compensated for the increase in risk imposed by asthma, so that the difference in risk between asthma and no asthma may have become less evident in older age groups.
RESPONSE: Thank you for this point. We now comment on the contrast between the finding of reference 8 and reference 15 on page 3, lines 127-136.
- 4) I suggest that the Authors move the paragraph on the association between asthma and coagulation abnormalities after all paragraphs on the association between asthma and clinical endpoints. That is, the lines 104 to 110 (from “Sneeboer et al.” to “but not with eosinophilic counts”) should be moved after line 136 and before the concluding paragraph (lines 137-144). In this way, the nice paragraph between 137-144 sums up the evidence just presented in the same order (as the “pathophysiological mechanisms” are presented last in the sentence 140-142).
RESPONSE: Thank you for this point. We have revised the paragraph accordingly.
- MINOR/FORMAL ISSUES. Would it be not possible to have the tables in the main text rather than as supplementary tables? It is normal for a large review to have several tables. I cannot find a limit to the number of tables in the main manuscript for Journal of Clinical Medicine. Of course, if the Editorial Office sets forth that they should be in the Supplement for space reasons, then keep them in the Supplement.
RESPONSE: Thank you for this point. In the revision we have added the tables in the main text.
- Page 4, line 137> “In conclusion” may replace “To sum”.
RESPONSE: Thank you for this point. We have revised the paragraph accordingly.
- Page 8, line 404: “Studies that were conducted pursuing VTE prevalence showed that patients with TB 404 are at increased risk.”. It would be simpler as “VTE prevalence is higher in patients with TB.”.
RESPONSE: Thank you for this comment. We have revised the sentence accordingly.
- Page 9, lines 446-447: The question that forms the title of section 6 is nice, but a little too long. Please shorten it by saving a couple words: “6. In situ thrombosis of the pulmonary arteries: another piece to the puzzle of acute VTE in patients with chronic inflammatory lung diseases?”
RESPONSE: Thank you for this suggestion. We have revised the title accordingly.
- Page 9 to 14, lines 456-457:“In pulmonary thrombosis, the embolization of a peripheral clot to the pulmonary arteries might not always be the case”. Maybe clearer if re-formulated as follows: “Pulmonary thrombosis may not always result from embolization of a peripheral clot to the pulmonary arteries”.
RESPONSE: Thank you for this point. We have revised the sentence accordingly (page 2, lines 54-55).
We found your feedback very constructive. We tried to be responsive to your concerns. We really thank you for taking the time and energy to help us improve this paper.